# OpenReview forum: "Faithful and Unfaithful Error Recovery in Chain of Thought"
_colmweb.org/COLM/2024/Conference — COLM_

### Official Review · Reviewer_kvJb · 2024-04-11

**Rating:** 4
**Confidence:** 4
**Ethics Flag:** 1

**Summary:**

This paper analyzes error recovery behaviors in chain-of-though reasoning generated by LLMs. Their experiments introduce artificial mistakes in numbers in chain-of-thought reasoning on arithmetic reasoning tasks and evaluate whether LLMs can recover from the errors (faithfully or unfaithfully) or not. This paper shows that (1) LLMs often can faithfully recover from errors with contexts that provide more evidence for the correct answer (e.g., copying errors), (2) correct recovery occurs more often on errors with greater magnitude, and (3) context noise and error recovery prompts make LLMs more careful about errors.

**Reasons To Accept:**

Analyzing mistakes in chain-of-thought reasoning is an important research topic. Although post-hoc detection or correction of errors has been widely studied, error recovery is a relatively less explored topic and is a reasonable direction for analyzing the behaviors of LLMs.

**Reasons To Reject:**

This paper does not provide sufficiently useful or novel observations.

I recommend targeting more realistic errors made by recent LLMs. (1) First, this paper only targets artificial mistakes in the numbers of arithmetic reasoning. (2) In addition, the proposed three types of errors (page 5) are not representative of mistakes made by recent LLMs. I believe that recent strong LLMs do not often make simple copying errors or calculation errors. Analysis of too simple errors that recent models will not make is not very useful. This paper should first analyze the actual mistakes made by LLMs.

---

> ### Author Rebuttal · Authors · 2024-05-31
>
> >I recommend targeting more realistic errors made by recent LLMs. (1) First, this paper only targets artificial mistakes in the numbers of arithmetic reasoning. (2) In addition, the proposed three types of errors (page 5) are not representative of mistakes made by recent LLMs. I believe that recent strong LLMs do not often make simple copying errors or calculation errors. Analysis of too simple errors that recent models will not make is not very useful. This paper should first analyze the actual mistakes made by LLMs.
>
> Thank you for your insightful feedback. We agree that analyzing more realistic errors made by recent LLMs is an important direction for future work. Our current study focused on artificial errors so that we can carefully control the experimental conditions, and eliminate any possible confounds. However, these may not fully capture the mistakes of state-of-the-art models in the wild.
>
> In the revised manuscript, we will expand our discussion of this limitation and highlight the need for future research to:
>
> 1. Characterize the types of errors current high-performing models actually make on various reasoning tasks.
> 2. Design controlled experiments to study recovery from these naturalistic error types.
>
> We believe our work provides a useful foundation for this future research.
>
> We also see value in studying artificial errors in more complex reasoning domains, as a bridge between our current work and future naturalistic error studies.

---

> > ### Comment · Reviewer_kvJb · 2024-06-03
> > **Re: Rebuttal by Authors**
> >
> > Thank you for your response.
> >
> > > Our current study focused on artificial errors so that we can carefully control the experimental conditions, and eliminate any possible confounds
> >
> > While this is a possible direction for analysis, I believe that artificial errors should try to mimic actual errors made by recent LLMs to provide useful observations.
> >
> > > 1. Characterize the types of errors current high-performing models actually make on various reasoning tasks.
> >
> > I agree this point should be added to your paper. I look forward to seeing the updated version.

---

> > > ### Author Response · Authors · 2024-06-05
> > > **Analysis of actual errors**
> > >
> > > >I believe that artificial errors should try to mimic actual errors made by recent LLMs to provide useful observations.
> > >
> > > We agree that it is important to know whether the errors in our experiment occur naturally. We have now examined a random sample of errors that GPT-4 makes on the four datasets in our paper. These are questions where GPT-4 did not generate the correct answer in its initial response, before we performed any of the artificial error interventions. These responses were previously discarded in our analyses.
> > >
> > > Out of 74 errors, 7 fit the definition of calculation errors provided in the paper. 6 of the other errors involve the model skipping intermediate reasoning steps, similar to the recovery behavior observed in the paper.
> > >
> > > The full annotated data is available at [this link](https://github.com/CoTErrorRecovery/CoTErrorRecovery/blob/main/results/error_audit.csv).
> > >
> > > We believe that these findings offer preliminary evidence that the errors introduced in the paper do occur naturally.

---

### Official Review · Reviewer_QfWH · 2024-05-11

**Rating:** 7
**Confidence:** 3
**Ethics Flag:** 1

**Summary:**

This paper studies error recovery in chain of thought, focusing mainly on GPT-4. When errors are introduced in its CoT generations, what kind of errors is it more likely to correct, and is it likely to correct them silently or point out the prior mistake? The paper presents the results of several experiments, varying the location & propagation of the error in the explanation, the magnitude of the error, and features in the input that may indicate to the LM an increased likelihood of errors.

The study is interesting and reasonable. The results should be heavily contingent on the way the model is trained in RLHF — it's easy to imagine small differences in the annotation methodology (i.e., do you train it to correct mistakes, and do you train it to point out such corrections? Think analogs of scheduled sampling, or whether perturbed/mistaken inputs are sampled during RLHF annotation) having big effects on the results. So I don't know how the results will generalize per se. However, the paper's point about faithful versus unfaithful recoveries is a good one, and seems interesting and relevant to study. Having this evaluation out there as a benchmark on frontier LMs seems pretty reasonable too, though requiring human annotation does make the situation a bit more difficult.

**Questions To Authors:**

Minor comments:
* Check for "Shapely values" typo (should be Shapley)
* On Page 4, you mention that you sample CoTs where the final answer was correct, "indicating that the chain of thought was correct". Doesn't this assume the models have a 0% error recovery rate by default? Or at least 0% unfaithful recovery. If this is true, then does that undermine the original motivation for the work? I'd at least like to see this assumption validated to see properly how much it's bent in the data.

EDIT: As a side note, I agree with Reviewer METV's comments on the use of the words "faithful" and "unfaithful" and also encourage the authors to adopt different terminology. I acknowledge that some previous work has used these terms to refer to the connection between the CoT explanation and answer rather than the model's decision process, but I think this created a lot of confusion and we should all try and stick with the latter.

**Reasons To Accept:**

The experiments are clever and strong.
* Measuring recovery rates for different kinds of errors (copying, calculation, etc.) is interesting and well-motivated.
* Looking at error magnitude is a more obvious choice but also a good one, and the effect on faithful/unfaithful recoveries is interesting.
* Prompting the model to expect mistakes is also something that definitely should've been tried, glad it was.

Overall it seems like a strong study of a focused phenomenon.

**Reasons To Reject:**

I feel like there is not as much content in the paper as I would've expected for a 9 page paper but I think it might just be the COLM format. And my MO for reviewing is to score based on soundness, so I won't dock the paper for this.

My biggest complaint is probably that I'm left with lots of questions about how the outcomes measured here vary with different model training methodologies. I feel it would've been more useful to do this on, e.g., an open-source model, so that follow-up studies would be able to measure the effect of new training regimes (e.g., to explicitly encourage faithful error recovery).

EDIT: I actually agree with Reviewer kvJb's criticism that the study only focuses on these artificial/constructed errors, which may not resemble LLM errors in the wild, and that it feels more immediately important and impactful to study the kinds of errors that LLMs actually make. But I feel that would require a pretty different kind of study than the current one, and wouldn't afford nearly as much control to study specific questions like the authors did in this case. So it seems to me like it'd just be a different paper, and that sort of study would be hard to fit in as an extra with the work already presented here. Perhaps a compromise would be to report some kind of aggregate of the likelihood ratios of your edits to the CoTs — i.e., when you insert the error, how does the likelihood of the CoT that you're prompting it with change from the original one? If the likelihood doesn't go down by much, that is some evidence that the mistake is "plausible" (i.e., could happen by sampling).

---

> ### Author Rebuttal · Authors · 2024-05-31
>
> **Open-source models:**
>
> Thank you for raising this important point. We agree that using open-source models would make it easier to investigate how different training methodologies affect the error recovery behaviors we have found.
>
> We did perform some initial experiments on Llama-2 70B. However, the baseline recovery rates for this model was too low to allow for meaningful statistical analysis of the effects of our experimental manipulations. For example, in the GSM8K calculation error condition, Llama-2's recovery rate was only 11%, compared to 80% for GPT-4.
>
> We hypothesize that the lower recovery rate in Llama-2 (and GPT-3.5) may be due to lower overall reasoning abilities in these models.
>
> In the revised version of the paper, we will present results for Llama-3, which may have higher recovery rates due to improved reasoning ability.
>
> **Natural vs. artificial errors:**
>
> We appreciate this feedback and agree that studying LLM errors in the wild is an important direction for future research. As you note, investigating naturalistic errors would likely require a different study design than the controlled error paradigm we employed here to probe specific questions about recovery behaviors.
>
> We believe both approaches have value. Our current study sacrifices some external validity for experimental control. Studying naturalistic LLM errors has more direct implications for applications, but limits the types of questions that can be asked about LLMs and their error recovery behavior.
>
> Regarding your suggestion to report the likelihood ratios of our edited CoTs compared to the original ones – this is a great idea for assessing the plausibility of our artificial errors. We will include this analysis in the revised version. If the likelihood does not substantially decrease, that would suggest our errors are reasonably plausible and could arise through the model's own sampling process.
>
> **Correctness of original CoT transcripts:**
>
> Thank you for pointing this out. This was incorrectly stated in the paper. We manually inspected the region surrounding the target error that we introduced, and filtered any CoT transcripts with incorrect reasoning in this region. Incorrect reasoning occurred in 1-13% of the CoT transcripts, depending on the dataset.

---

### Official Review · Reviewer_kP29 · 2024-05-11

**Rating:** 7
**Confidence:** 4
**Ethics Flag:** 1

**Summary:**

The paper investigates the faithfulness of error recovery behaviors in large language models (LLMs) during Chain of Thought reasoning. The authors analyze the factors that influence error recovery, such as the amount of evidence for the correct answer and the magnitude of the error. They also examine the impact of prior expectations on error recovery. The study finds that error recovery is influenced by these factors and that there is a dissociation between faithful and unfaithful error recoveries.

**Questions To Authors:**

(1) I was wondering whether the findings in this paper could be generalized to open-source / smaller models? For example,  could the authors add some experiments with llama 3 series in different model sizes?

(2) Previous work has investigated how to make LLMs generate more effective chains of thought, incorporating aspects like selection [1], diversity [2], uncertainty [3], and difficulty [4]. I am very interested in how the author's findings in this paper can be applied to enhance the model's reasoning capabilities. Could the author possibly discuss these works to provide some outlook?

[1] Automatic Prompt Augmentation and Selection with Chain-of-Thought from Labeled Data https://arxiv.org/abs/2302.12822

[2] Automatic Chain of Thought Prompting in Large Language Models https://arxiv.org/abs/2210.03493

[3] Active Prompting with Chain-of-Thought for Large Language Models https://arxiv.org/abs/2302.12246

[4] Complexity-Based Prompting for Multi-step ReasoningDownload https://openreview.net/forum?id=yf1icZHC-l9

**Reasons To Accept:**

(1) The paper provides a comprehensive analysis of error recovery behaviors in LLMs during Chain of Thought reasoning.

(2) The experiments are well-designed and provide insights into the factors that influence error recovery.

(3) The paper contributes to the understanding of the faithfulness of LLM reasoning and highlights the dissociation between faithful and unfaithful error recoveries.

**Reasons To Reject:**

This is a very insightful paper. Actually I do not see any reasons to reject while I have some questions to authors. Please see Questions To Authors

---

> ### Author Rebuttal · Authors · 2024-05-31
>
> > (1) I was wondering whether the findings in this paper could be generalized to open-source / smaller models? For example, could the authors add some experiments with llama 3 series in different model sizes?
>
> We performed experiments on Llama 2 70B, but the recovery rates were too low to allow for meaningful analysis. We will perform experiments with Llama 3 for the revised version.
>
> >(2) Previous work has investigated how to make LLMs generate more effective chains of thought, incorporating aspects like selection [1], diversity [2], uncertainty [3], and difficulty [4]. I am very interested in how the author's findings in this paper can be applied to enhance the model's reasoning capabilities. Could the author possibly discuss these works to provide some outlook?
>
> Thank you for providing these relevant citations on improving LLM reasoning. We see several connections between these papers and our findings on error recovery.
>
> The uncertainty-based methods in [1] and [3] could potentially identify examples where error recovery is most important. The complexity-based prompting from [4] is related to our observation that models recover more effectively when expecting errors in a problem. The diversity approaches in [2] are related to our findings on the effects of varied error types and magnitudes on the rate of recovery.
>
> We appreciate you pointing us to these works and will incorporate a discussion of these connections in the revised manuscript.

---

> > ### Comment · Reviewer_kP29 · 2024-06-05
> >
> > Thank you for the response. It is a pity not to see generalization to open-source models (llama-2) but I agree with you that trying with llama-3 is worthwhile to do since llama-3 is more powerful than llama-2 both from their tech report and from my experience.
> >
> > I will keep my rating unchanged.

---

### Official Review · Reviewer_METV · 2024-05-11

**Rating:** 7
**Confidence:** 4
**Ethics Flag:** 1

**Summary:**

This paper analyzes the effects on **GPT-4 (0314)** and **GPT-3.5**’s ability to perform **error recovery**—make the right prediction despite making a wrong reseasoning step when chain-of-thought prompted—in the context of **math word problems**.

It separates so-called **“faithful”** error recovery—when a model generates language that acknowledges a mistake was made and corrects it in the subsequent token generations; for example:
> “Oops, I made a mistake! 3 spiders have 3 x 8 = 24 legs.”,

from **”unfaithful’”** error recovery—when the error is not acknowledged and explicitly corrected. I have concerns regarding using the term “faithful” to describe these behaviors and I strongly urge the author to re-consider this choice (see “Questions To Authors”).

To perform the analysis, this paper first generates chain-of-though reasoning. If the answer is correct, three types of errors are introduced and it’s expected that it’s easier to recover from some errors than others. Given the text up to the error and the error, the LLM is prompted again. Then, it is manually analyzed whether there is clear evidence for error recovery or not (faithful vs. unfaithful), relative to the error type. Additionally, the effect of the difference between the correct and erroneous values, as well as the effect of prompting the model to expect errors, is analyzed.

**The outcomes of the analysis align with expectations**: harder errors are harder to recover from, larger differences between the correct and erroneous values result in a model more faithfully (read: transparently) recovering its errors, and if the model is “warned” the errors might happen, it recovers from errors better.

**I support such confirmatory analysis given the numerous paradoxical behaviors exhibited by large-scale models.**

**Questions To Authors:**

- **Clarification question 1: the correct answer indicates the original CoT was correct.** There is evidence that CoT may not be correct although the answer is correct (e.g. https://arxiv.org/abs/2402.16048v1). What is written under Table 1 seems to then be making a wrong assumption. Please clarify this.

- **Clarification question 2: why do faithful recovery and unfaithful recovery do not sum to 100%**. My understanding is that faithful recovery is `count(**clear** evidence of error recovery)/count(correct final answer)` and unfaithful recovery is `count(**unclear** evidence of error recovery)/count(correct final answer)`.

---

Suggestions for improvement:


- **Major: On using the term “faithful”.** If the model directly re-does calculation without explicitly identifying the error, or states the correct value after the error, does not exclude the possibility that the model did this internally but did not spell this out. This is also an issue when this term is used for faithfulness evaluations that intervene on the reasoningm such as introducing errors: the model might ignore the error in the generated reasoning, use the correct reasoning internally, and come to the right answer. I strongly urge you to reconsider describing “(un)faithful error recovery” as “(un)acknowledged” or “(un)transparent” or “(un)clear” or “(un)supported”.

- **Major: On connections between error recovery and CoT faithfulness.** This paper connects error recovery with prior literature that assumes that if a model reaches a correct answer with wrong reasoning, then the reasoning is unfaithful to the model, i.e., the reasoning is disassociated with the internal computations that were done to generate the answer. I find this assumption too strong, and I personally think that this work does not need to situate itself relative to this assumption and that studying this topic on its own is valuable and interesting. A system that generates correct answers preceded by wrong reasoning in natural language obviously won’t be helpful to laypeople. I am making this point because the connections made are vague to me, such as: “We challenge the assumption that error recovery indicates unfaithful reasoning.” and “The current paper finds that it is easy to induce unfaithful reasoning in LLMs by forcing error recovery.”

- **Explanations for Figure 4**: Why is the total error recovery and propagated calculation error notably smaller for MultiArith and GSM85 than other datasets?

- You mention the dissociation paradigm for psychology in the introduction, but this is mentioned again next in the discussion in Conclusions. I urge you to omit this because your analysis methodology does not seem motivated by psychology since it's pretty straightforward, so this felt misleading to me.

Writing improvements:

- Abstract: "LLMs improve their performance in..." This is not generally true and you should weaken this. For example, in the clinical domain this paper shows https://arxiv.org/abs/2212.13138
> Somewhat unexpectedly, we did not observe improvements using CoT over the standard few-shot prompting strategy across the three
multiple-choice datasets - MedQA, MedMCQA and PubMedQA.

- All mention of "faithful" error recovery on the first page is vague because you did not define it anywhere.

- In Section 2.2.: "naturally produce errors" I am confused by "naturally" here. What would be "unnatural" error production?

- You should add works that study the faithfulness of free-text or natural language explanations which is another term people use in place of chain-of-thoughts. For example, https://aclanthology.org/2023.acl-short.25/, https://arxiv.org/pdf/2311.07466, https://aclanthology.org/2021.emnlp-main.804/.

- The intro of Section 3 says that error recovery was defined in 2.2, but it's not explicitly defined and is mentioned only later in 2.2. I recommend that you say explicitly what error recovery is in Section 3.

- It would be nice to immediately illustrate at the end of page 2 why the assumption in prior work might not hold.

- On the top of page 4, when you say "ground-truth CoT transcripts" it is unclear ground-truth for what.

- Similarly, it's unclear in "We evaluated fixed versions of GPT-3.5..." in 4.1.1 what you are evaluating.

- It's strange to me to have subsections for 2-3 sentences.

- "See Appendix B.1..." is repeated.

- Given my question of why the faithful and unfaithful recovery do not sum to 100, you should clearly define these measurements.

- "...errors by changing their magnitude" was unclear to me until I read the last paragraph in the intro of Section 6. This should be clear immediately.

- The label of y-axis in Figure 6 is wrong.

- The difference for copy error in Figure 7 does not seem significant. I recommend breaking the significance testing by error type if it's not already.

- I had a hard time following the discussion in Conclusions and found the connection with cognitive processes too shallow and unnecessary.

**Reasons To Accept:**

Enabling prompting models to self-correct transparently is important for LLM applications where the reasoning is presented to people. The analysis presented in this paper moves us toward that direction.

Specifically, from this paper we learn that we should focus on developing methods to recover from the harder error types and cases where the difference between the correct and erroneous values is smaller, as well as use the prompt that “warns” that the errors might happen.

**Reasons To Reject:**

I do not notice major issues with the soundness of this work that would make me argue against accepting it. That said, I do have two soundness-related questions that I hope the authors can clarify (see “Questions To Authors”) in a way that won't change my soundness perception.

The analysis is not as comprehensive as strong analyses published in a top-venues; specifically:

- Only reasoning for math word problems is analyzed. Due to this, adding, controlling, and evaluating errors is easy relative to what would need to be done for other types of reasoning (e.g., commonsense).

- 300 instances are analyzed.

- A single-family of proprietary models is used (GPT).

- The effecting variables are straightforward to come up with.

- Some effects do not apply to other reasoning types, e.g., magnitude.

Finally, there are some issues with terminology, positioning, and argumentation that I would like to see improved (Questions To Authors), but none of them are deeply flawed (or more than a typical LLM paper these days).

---

> ### Author Rebuttal · Authors · 2024-05-31
>
> **Clarification question 1:**
>
> This was incorrectly stated in Table 1. We manually inspected the region surrounding the target error that we introduced, and filtered any CoT transcripts with incorrect reasoning in this region. Incorrect reasoning occurred in 1-13% of the CoT transcripts, depending on the dataset.
>
> **Clarification question 2:**
>
> Faithful recovery and unfaithful recovery are reported as absolute quantities, not as a proportion of all recoveries. 1 - (faithful + unfaithful) is the rate of non-recovery responses.
>
> **Comprehensiveness of analysis:**
>
> We agree that extending this analysis to other reasoning domains beyond math word problems is an important direction for future work. While introducing controlled errors is most straightforward for numerical problems, we believe similar methodologies could be adapted for other tasks, such as using semantic similarity to define larger vs smaller errors. We will note this as an important area for follow-up research.
>
> Regarding the sample size, 300 instances are being evaluated per dataset. This number was chosen based on a statistical power analysis. Given the large effect sizes observed, our analyses were significantly overpowered, and fewer samples likely would have sufficed. We will clarify this rationale in the revision.
>
> For model diversity, we focused on GPT-4 because it was the strongest model available at the time of submission. Preliminary experiments with Llama-2 70B yielded recovery rates too low for meaningful analysis. However, we agree this is a limitation and will include results for Claude Opus in the revision.
>
> **Terminology and framing:**
>
> We appreciate the suggestion to reconsider the "faithful" terminology, as we agree it implies stronger claims about the model's internal reasoning than we can confidently make. We primarily used it to connect to prior work, but given your feedback to reframe the positioning of the paper, we will adopt alternative terms like "(un)acknowledged" or "(un)supported" recovery.
>
> Thank you also for suggesting that our work can stand on its own in studying error recovery behavior. Those specific quotations are related to previous work, for example https://arxiv.org/abs/2307.13702, which identifies all error recoveries as unfaithful. We will revise the framing, clarifying the remarks on prior work while focusing more on the recovery phenomenon itself.

---

> > ### Comment · Reviewer_METV · 2024-06-03
> > **Re: CoT correctness & Recovery measurements**
> >
> > Thank you for your clarifications. I'm glad to hear CoT correctness was determined manually. Please make that clear in the paper too.
> >
> > Regarding this:
> >
> > > Faithful recovery and unfaithful recovery are reported as absolute quantities, not as a proportion of all recoveries. 1 - (faithful + unfaithful) is the rate of non-recovery responses.
> >
> > You label y-axis with "Faithful Recovery **(%)**" and "Unfaithful Recovery **(%)**" (emphasis mine). Using the percentage symbols for absolute quantities, especially when the total count per error is 100, is misleading. (Also in Table 2). The absolute values also make it hard to compare these faithful recovery rates across error types. This should be improved.

---

> > > ### Author Response · Authors · 2024-06-03
> > > **Terminology clarification**
> > >
> > > Apologies for our confusing terminology. The y-axis indicates the rate of faithful/unfaithful recovery as a proportion of **all** responses (not just recovery responses). So it is indeed a percentage. This is why 1 - (Faithful Recovery (%) + Unfaithful Recovery (%)) is the rate of non-recovery responses.

---

> > > > ### Comment · Reviewer_METV · 2024-06-04
> > > > **Re: Terminolog  / Recovery measurements**
> > > >
> > > > The proportion of all responses should not be described as "**absolute** quantities" as in your previous response.
> > > >
> > > > I understand what you are calculating now. I still recommend using the total recovery count for normalization to make it easier to compare faithful recovery across different error types with varying recovery counts.

---

### Decision · Program_Chairs · 2024-07-10

**Decision:**

Accept

**Comment:**

The authors investigate the behavior of LLMs in correcting its own decisions made in a chain of thoughts. Artificial mistakes are introduced so that controlled testing can be done. The experiments are invasively designed and the conclusion gives some insights on the area that has not been intensively discussed in the literature.

Reviewers questioned the notations, the generality in terms of the LLMs chosen, and the correlation between the artificial errors and real mistakes that are made by LLMs. The authors engaged the reviewers in multiple rounds of discussions, which I enjoyed to read. Overall, I think the authors address most of the concerns. Despite the fact that the submission needs much revision to correct all the issues (where the reviewing process does not allow uploading revision), there is reason to foresee necessary changes being integrated.

[comments from the PCs] Please follow up on the revisions noted by the AC.